# Exploring the Diversity of Biofilm Formation by the Food Spoiler *Brochothrix thermosphacta*

**DOI:** 10.3390/microorganisms10122474

**Published:** 2022-12-15

**Authors:** Antoine Gaillac, Romain Briandet, Elodie Delahaye, Julien Deschamps, Evelyne Vigneau, Philippe Courcoux, Emmanuel Jaffrès, Hervé Prévost

**Affiliations:** 1Oniris, INRAE, Secalim, 44300 Nantes, France; 2INRAE, AgroParisTech, Micalis Institute, Université Paris-Saclay, 70123 Jouy-en-Josas, France; 3Oniris, StatSC, 44300 Nantes, France

**Keywords:** *Brochothrix thermosphacta*, biofilm, biofilm ring test, crystal violet, confocal laser scanning microscopy, diversity

## Abstract

*Brochothrix thermosphacta* is considered as a major spoiler of meat and seafood products. This study explores the biofilm formation ability and the biofilm structural diversity of 30 multi-origin *B. thermosphacta* strains using a set of complementary biofilm assays (biofilm ring test, crystal violet staining, and confocal laser scanning microscopy). Two major groups corresponding to low and high biofilm producers were identified. High biofilm producers presented flat architectures characterized by high surface coverage, high cell biovolume, and high surface area.

## 1. Introduction

*Brochothrix thermosphacta* is a Gram-positive, rod-shaped, non-pathogenic, psychrotrophic bacterium, considered as one of the main spoilers of seafood and meat products [1,2]. *B. thermosphacta* has been isolated from a wide range of spoiled foods, such as poultry, beef, pork, cheese, fish, and seafood [3,4]. It can produce volatile organic off-odor compounds leading to food waste, which contributes to the worldwide economic and ecological impact of food spoilage [5,6]. In seafood products, such as cold-smoked salmon, *B. thermosphacta* can produce butter/plastic/rancid, blue cheese, sour/pungent off-odors [7,8,9] and strong butter, buttermilk-like, sour, and nauseous off-odors in cooked and peeled shrimp [10,11,12]. It can also produce cheesy and creamy dairy off-odors in beef [13,14].

*B. thermosphacta* is widely disseminated along the food processing chain, from the raw material to the final product. It has also been isolated from food processing plants (e.g., floors, walls, machines) and is considered as one of the main sources of food bacterial contamination during processing [2,15,16,17]. Many of these studies suggest that *B. thermosphacta* could be persistent on these surfaces thanks to its ability to form biofilms. However, to our knowledge only three papers describe biofilm formation by *B. thermosphacta* in a laboratory model [18,19,20]. *B. thermosphacta* was recently identified in 80% of the biofilms collected in a meat processing plant, including food contact and non-food contact surfaces. A strain isolated from those industrial biofilms was classified as a low biofilm producer based on cell density in the mature biofilm, the amount of carbohydrates in the matrix, and the number of genes known to be associated with biofilm formation [21,22]. A genomic comparison of six *B. thermosphacta* strains identified a putative O-acetyltransferase similar to *Bacillus subtilis* O-acetyltransferase (EpsM) and involved in biofilm matrix production [23]. The gene encoding the putative EpsM protein was present in the genome of three strains, likely indicating a diversity of biofilm traits within this species. Genes involved in cellulose production were identified in *B. thermosphacta* genomes [22]. Cellulose production genes were already associated with biofilm matrix production in *Salmonella enterica* [24].

Moreover, two types of cell morphology were identified, type I (BI-like, fluffy-cloud-like clumps) and type II (BII-like dense clusters with protrusions); these morphotypes may influence cell aggregation and biofilm formation [25]. However, to our knowledge, no study on the biofilm-forming ability on abiotic surfaces with so many different strains of *B. thermosphacta* has been published yet.

A biofilm is a three-dimensional microbial community associated with a surface and embedded in a self-produced extracellular matrix, mainly composed of polysaccharides, nucleic acids, and proteins [26,27]. Biofilm architecture is known to affect molecular diffusion/reaction of nutrients and antimicrobial agents; protect cells against desiccation and mechanical stresses; and trigger cell-to-cell communication, genetic plasticity, and cell type diversification [28].

Numerous complementary methods have been developed for in vitro biofilm growth and quantification but without any international standardization to date [29,30]. Biofilms reactor devices have been developed for bacterial biofilm culture with established standard assays by the American Society for Testing and Material [31]. The crystal violet (CV) staining methods in microplates and its variant were widely used to quantify fixed biomass. CV methods were adapted and improved from various microorganisms, including human pathogens [32,33,34,35,36] and food-processing-associated microorganisms [37,38]. However, despite this method having the potential to be poorly reproducible, it is well adapted to compare the biofilm phenotype of a wild type (WT) strain with collections of isogenic mutants affected in candidate biofilm determinants [39]. It can be more difficult to interpret in the presence of collections of natural isolates that can react differentially with CV dyes. Moreover, it has a low sensitivity that precludes quantitative analysis of the early stages of biofilm formation. Di Domenico et al. [33,40] developed the clinical biofilm ring test (cBRT), adapted from the BioFilm Ring Test (BRT) to evaluate the initiation of biofilm formation in a large range of bacterial species. The cBRT is based on the immobilization of magnetic microbeads within the biofilm matrix, during growth in microplate wells [41]. It is rapid and robust to evaluating microbial biofilm initiation and is suitable for high-throughput screening in clinical microbiology [33,40]. Biofilm architecture has been mostly investigated by confocal laser scanning microscopy (CLSM) and derived fluorescent microscopy tools. Geometric quantitative biofilm parameters can be extracted from CLSM images with different computational tools, such as PHLIP, COMSTAT, or IMARIS [42]. BiofilmQ—an advanced biofilm analysis tool—was recently developed to analyze fluorescence images of spatially organized microbial communities [43]. From biofilm images, this software can extract conventional geometric descriptors and dissect the biofilm biovolume into a cubical grid, with a user-defined cube size. Based on internal parameters extraction from each cube of the segmented biofilm, BiofilmQ can extract 420 global and internal biofilm parameters from CLSM images.

This article explores the production capability and structure diversity of *B. thermosphacta* biofilms. This is important to improve our knowledge on biofilm behaviour in the food processing environment and to better control food contamination by this bacteria. In this study, 30 *B. thermosphacta* strains isolated from diverse origins were investigated to evaluate biofilm initiation and formation using the cBRT and CV staining. The 3D structure of *B. thermosphacta* biofilms was deciphered by CLSM, combined with the BiofilmQ image analysis toolbox. To our knowledge, this study is the first to describe the diversity of *B. thermosphacta* biofilm formation with a set of complementary biofilm assays on an abiotic surface.

## 2. Materials and Methods

### 2.1. Brochothrix thermosphacta Strains and Culture Conditions

Thirty *B. thermosphacta* strains were used (Table 1), mainly selected from our laboratory collection (INRAE-Secalim, Paris, France), IFREMER Nantes *B. thermosphacta* collection, and two were selected from the DSMZ-German Collection of Microorganisms and Cell Cultures (DSMZ 20171T and 20599). The main selection criterion was the diversity of environmental sources (food matrices or abiotic surfaces). *B. thermosphacta* strains were grown at 25 °C in brain heart infusion (BHI) medium (VWR chemicals, France) from a working stock prepared in BHI broth with 20% glycerol (VWR chemicals, France), and then stored at −80 °C. The bacteria were subcultured on BHI-agar plates at 25 °C for 24 h before use.

### 2.2. Assessment of the Biofilm-Forming Potential of B.thermosphacta Isolates Using the Biofilm Ring Test

The cBRT (KitC004, BioFilm Control, Paris, France) was carried out in polystyrene 96-well microplates, as described by Di Domenico et al. (2016) [33]. Briefly, BHI broth from the BRT kit was supplemented with toner containing magnetic beads (Toner4 Biofilm Control, Paris, France) at a final concentration of 10 µL/mL (BHI + T4). A 24 h BHI agar plate culture of each *B. thermosphacta* strain was used to prepare the S1 cell suspension (2 × 10^6^ CFU in 1 mL of BHI + T4). An amount of 200 µL of BHI + T4 were transferred in each well of a microplate. An amount of 200 µL of the S1 cell suspension were added (3 wells per strain, 10^6^ CFUs/mL), and then a 1/2 serial dilution was applied to reach a 1/64 dilution (1.6 × 10^4^ CFUs/mL). The non-inoculated wells containing only BHI + T4 were used as negative controls. The microplates were incubated statically at 25 °C. After 4 h of incubation, 100 µL of contrasting agent (LIC001, inert opaque oil) were added, and the plates were placed on the test block to apply a local magnetic field to the centre of each well for 1 min. The magnetic field attracts the free beads to the centre of the well, which form a dense brown spot made of aggregated particles. The plate was scanned with the plate reader (Pack BIOFILM, Biofilm Control, Paris, France). The images were analyzed with BioFilm Control Elements^®^ 3.0 software to obtain a numerical value called ‘biofilm index’ (BFI), ranging from 0 to 20 for each well. In the absence of biofilm formation by bacteria, the beads are highly mobile and easily reach the centre of the well, hence a high BFI. Conversely, a low or null BFI is obtained when the beads are immobilized within a biofilm matrix. The BFI values were used to measure the biofilm-forming potential (BP), as described by di Domenico et al. (2016) [33]. The BP was calculated for each well and each plate using the following formula:

BP = [1 − (BFI sample/average BFI of negative control)].

The average BFI of the negative control was determined for each plate ranging from 18 to 20. If the BFI of a studied strain was equal to the BFI of the negative control, the BP of the strain was considered null (0). In contrast, if the BFI value of a strain was low or null, the BP could reach the maximum value (1). In order to classify the ability of the strains to initiate biofilm formation, the average BP was calculated for each bacterial concentration and compared with the specific cut-offs (BPc), determined from the following formula:

[1 − ((average BFIs of negative control − three standard deviation)/2)/average BFI of negative control] [47].

The ability of the strains to initiate biofilm formation was determined using the minimum cell concentration when the BP was higher than the BPc. The concentration interval ranged between 1.00 × 10^6^ CFU/mL for poor early biofilm producers [33] and 1.56 × 10^4^ CFU/mL because none of our 30 strains reached the BPc at this concentration. The strains were classified into the following categories: poor biofilm producer (10^6^ CFU/mL), weak biofilm producer (5.00 × 10^5^ to 2.50 × 10^5^ CFU/mL), moderate biofilm producer (1.25 × 10^5^ to 6.50 × 10^4^ CFU/mL), and high biofilm producer (3.13 × 10^4^ to 1.56 × 10^4^ CFU/mL). The experiments were performed in triplicate with independent *B. thermosphacta* cultures.

### 2.3. Biofilm Quantification Using the Crystal Violet Assay

The microtiter plate crystal violet assay was performed, as previously described [39], with slight modifications. Briefly, each strain was grown in BHI broth at 25 °C under shaking (150 rpm) overnight. The overnight culture was diluted 1/100, and 200 µL of this cell culture were transferred (8 wells per strain) in a 96-well polystyrene microplate (Thermo-scientific^®^ nunclon delta surface, Paris, France). Sterile BHI medium was used as a negative control. The microplates were incubated at 25 °C in static conditions. After 24 h, the medium and the suspended planktonic cells were discarded. To remove loosely attached bacteria, each well was washed three times with 200 µL of sterile water. For staining bacterial biomass, 200 µL of 0.1% crystal violet (CV) solution (Sigma Aldrich, 1% crystal violet) were added to each well, and then the microplate was incubated at room temperature for 20 min. Then, the CV solution was discarded and the wells were washed three times with 200 µL of sterile water. All the wells were filled with 200 µL of 96% ethanol and their content was homogenized by pipetting to completely dissolve biofilm-bound CV. The amount of destained CV was assayed by reading optical density (OD) at 570 nm in a microplate reader (TECAN Spark, Paris, France). Each experiment was performed in triplicate with independent *B. thermosphacta* cultures (24 measurements per strain).

### 2.4. Biofilm Structural Analysis by Microscopy

The 3D architecture and the spatial distribution of the biofilms formed by the thirty *B. thermosphacta* strains were examined by confocal laser scanning microscopy (CLSM). In the same way as in the CV staining assay, each strain was grown in BHI broth at 25 °C under shaking (150 rpm) overnight. The overnight culture was diluted 1/100 with BHI broth, then 200 µL of this cell culture dilution was transferred to 96-well polystyrene microtiter plates with a µClear base (Greiner Bio-One, Les Ulis, France). The microplates were incubated at 25 °C in static conditions. After 2 h (early biofilm) or 24 h (mature biofilm), the biofilms were labeled with Syto 9 (5 µM), a cell-permeant nucleic acid maker. Images were acquired on a Leica SP8 equipped with a ×63 APO water objective at the MIMA2 microscopy platform (https://www6.jouy.inrae.fr/mima2, accessed on 12 October 2022). The Syto 9-labeled bacteria were excited at 488 nm, and the emitted fluorescence was collected in the 500–550 nm range. Two fields of each well were scanned with a z-step of 1 µm, and each strain was grown in three wells considered as independent technical replicates. Two independent biological replicates were performed for a total of 12 z-image-series per strain. BiofilmQ [43] was used to extract quantitative biofilm parameters (Appendix A).

### 2.5. Statistical Analyses

Statistical analyses were performed using R-studio software (R4.1.1) on the Migale platform [48]. CV biofilm quantifications associated with OD_570_ measurements were compared with a multifactor analysis of variance (ANOVA) and a post hoc pairwise test plus a boxplot representation to highlight statistical biofilm production groups. To highlight structural diversity, 37 parameters extracted from CLSM images with BiofilmQ were used for statistical analyses. The parameters were analyzed using a principal component analysis (PCA), and the strains were clustered using hierarchical clustering on principal components (HCPC, available in FactoMineR R package). Segmentation of strains into groups was represented with fviz_dendro and fviz_cluster (available in FactoExtra R package). The parameters that most significantly contributed to the diversity identified by the PCA were analyzed using ANOVA plus a boxplot.

## 3. Results

### 3.1. Assessment of the Biofilm Production Capacity

The early biofilm formation ability of the 30 *B. thermosphacta* strains included in this study was investigated using the cBRT after growth at 25 °C for 4 h. The strains were grouped in four distinct biofilm initiation profiles, namely poor, weak, moderate, and high biofilm producers. Thirteen strains did not reach the BPc at 1 × 10^6^ CFU/mL and six reached the BPc for a minimum concentration of 1 × 10^6^ CFU/mL; they were all classified as poor early biofilm producers. Twenty-three *B. thermosphacta* strains were classified as poor (63%) or weak (13%) early biofilm producers (Figure 1). The two reference strains DSMZ 20171T and 20,599 were classified as poor and weak early biofilm producers, respectively. Five strains (17%) were classified as moderate early biofilm producers. Only two strains (CD337(2) and CD326(1); 7%) blocked bead aggregation at the lowest bacterial dilutions, corresponding to the high early biofilm producer category. These results highlight the intra-species diversity of the early biofilm production ability of *B. thermosphacta* strains. Only 7% of the strains of our collection appeared to be early high biofilm producers.

The mature biofilm formation ability of each strain was quantified using the CV staining method [39] after growth at 25 °C in static conditions for 24 h. Seven strains were grouped in five mature biofilm producer groups. The 23 remaining strains were considered as low mature biofilm producers (Figure 2). The seven high mature biofilm producer strains had been identified as high or medium early biofilm producers by the cBRT. The very poor biofilm producers identified by CV had been identified as weak and poor biofilm producers by the cBRT. These results show that 23.3% of the strains were high biofilm producers in our conditions, while 76.7% were low biofilm producers.

### 3.2. Analysis of the Biofilm 3D Structure

Section projections from CLSM image series of each strain observed after 2 and 24 h were extracted using IMARIS software. At the early biofilm stage (t = 2 h), the numbers of microcolonies and their height were very heterogeneous, suggesting strain specificities. However, at the mature biofilm stage (t = 24 h), *B. thermosphacta* architectures were flat, with low visible diversity between strains’ structural feature. To quantitatively analyze these architectures, 37 biofilm structural parameters were extracted with BiofilmQ software for each time point (74 parameters in total), and a hierarchical classification on principal components (HCPC) was used to cluster the 30 strains in 5 major structural groups (Figure 3).

Each biofilm from each strain was also represented in IMARIS projection from CLSM images at t = 2 h and t = 24 h, and they were grouped based on their PCA cluster origin (Figure 4). The first cluster was composed of three strains already classified as medium or high early biofilm producers by the cBRT and high mature biofilm producers by CV staining. The second cluster was composed of three medium–high early biofilm producers and high mature biofilm producers plus four poor–weak early biofilm producers and low mature biofilm producers (SF 779, BSBS1.6; BSBS1.3; BSBS2.3). In the third cluster, three strains were classified as poor–weak early biofilm producers and low mature biofilm producers, and one strain (19/R/633) was classified as a medium early biofilm producer by the cBRT and as a high mature biofilm producer by CV staining. The fourth cluster was composed of two strains classified as poor–weak early biofilm producers and low mature biofilm producers. Finally, cluster 5 was composed of 14 poor–weak early biofilm producers and low mature biofilm producers.

Quantitative structural information was extracted from CLSM image using BiofilmQ. At t = 2 h (Figure 4A), the images of cluster 1 biofilms showed the highest numbers of microcolonies and cells, while those of cluster 4 biofilms showed the lowest number of microcolonies and cells. The images of the other biofilm clusters showed variable amounts of microcolonies and cells depending on the strain. At t = 24 h (Figure 4B), no architectural difference was identified between the different clusters, except cluster 4, which had a lower biofilm height visible in the biofilm projection images.

## 4. Discussion

This study explores the formation and structural diversity of biofilms composed of *B. thermosphacta* isolates. Based on the presence of this bacterial species in different food environments [21,22] and on its genomic heterogeneity in terms of the presence of a gene potentially involved in biofilm matrix formation [23], we envisioned a possible diversity of *B. thermosphacta* biofilm formation and architecture. To explore this diversity, appropriate methods are needed to analyze the biofilm and its architecture during the different stages of biofilm formation. Biofilm formation by 30 *B. thermosphacta* strains was quantified at the biofilm initiation stage (4 h) by the cBRT and at the mature biofilm stage (24 h) by CV staining. Biofilm structural diversity was explored at the initiation and mature biofilm stages using CLSM and image analysis. Twenty-three isolates out of thirty (77%) were classified as poor or weak early biofilm producers by the cBRT and as low mature biofilm producers by CV staining. In contrast, the seven remaining isolates (23%) were classified as medium or high, early and mature biofilm producers. The correlation between the cBRT and CV methods has already been observed for other bacterial species [33,40,41,49]. A high proportion of low biofilm producers has also been observed in other microbial species. Among 38 *Escherichia coli* sp. strains, only 3 were classified as strongly adherent and 8 as moderately adherent [50]. The same proportion was observed for multi-species strains isolated from milk-processing surfaces in a dairy plant: 49% were non-biofilm producers, 20% were weak biofilm producers, 27% were moderate biofilm producers, and 3.5% were strong biofilm producers [51].

The high-throughput biofilm phenotypes of many species have been studied with CV assays. Eleven “well-defined biofilm production ability reference strains” [33], including *Pseudomonas aeruginosa*, *Klebsiella pneumoniae*, *Ralstonia mannitolilytica*, *Staphylococcus aureus*, and *Staphylococcus epidermidis*, were studied at 37 °C, following the same protocol as in the present study. None of these strains reached an OD = 2, whereas the *B. thermosphacta* biofilm producer strains of the present study showed a minimum OD of 2. Moreover, 143 *Listeria monocytogenes* strains cultured at 30 °C and 20 °C and 40 *L. monocytogenes* strains cultured at 37 °C for 72 h and stained with CV showed an OD < 2 [34,35]. This suggests a strong ability of a few *B. thermosphacta* isolates to produce biofilms compared to previously characterized bacterial species.

We observed the presence of biofilms at the mature stage by CLSM for all 30 strains, even for those characterized as low biofilm producers by cBRT and CV. BiofilmQ analysis of the structural biofilm parameters extracted from CLSM images clustered the strains in five groups separated by the first four dimensions of the PCA (Appendix A). The differences between the groups can be explained by any of these dimensions and their respective most influential parameters (Appendix A). For example, according to the first dimension, the strains included in groups 1 and 2 showed equivalent biofilm structures made of similar numbers of cells and similar surface areas, biofilm thickness, and biofilm volume at t = 24 h. Conversely, according to dimension 2, the strains included in groups 1 and 2 showed different biofilm structures, with different biofilm densities and roughness; surface areas per volume at t = 24 h; and different biofilm volumes, heights, thickness, surface areas, and roughness at t = 2 h.

These five groups showed differences in the number of cells in each biofilm, in biofilm thickness, volume, local density, outer surface, surface local roughness, outer surface per substrate, and local substrate area at t = 24 h. At t = 2 h, those groups showed differences in their biofilm surface per substrate area, shape volume, biofilm height, biofilm local density, biofilm thickness, and shape roughness. Some of these parameters have been used to describe biofilm architecture from CLSM images with other tools, such as MATLAB, PHLIP, or FIJI [42,52,53]. The large number of parameters extracted from the CLSM images with BiofilmQ allowed us to deeply characterize the differences between *B. thermosphacta* strains.

The most contributory parameters of each dimension were used to quantify the differences between the five clusters. The highest biofilm parameter values were identified in groups 1 and 2 (Appendix A). These clusters were composed of seven high biofilm-producing strains and three low biofilm-producing strains (cBRT/CV). To explain the presence of high and low biofilm producers in the same cluster, we hypothesized an effect of the washing step in the CV experiment, not present in the non-invasive CLSM experiment. As described by Azeredo and colleagues, the washing steps could affect the cohesion of the biofilm [30]. The low biofilm producers identified by CV staining and classified in the second cluster by CLSM could probably have formed a mature biofilm like the high biofilm producers did (cBRT/CV/CLSM), but their poorly cohesive structure exposed them to detachment during the intensive washing steps. The impact of washing methods on biofilms has been studied on *S. aureus* (SH1000), *S. epidermidis* (ATCC 12228), *S. carnosus* (TM300), *P. aeruginosa* (PA01), and *E. coli* (TG1) biofilms. The pipetting/washing method showed a higher rate of cell detachment and affected the biofilm structure [32]. Based on all the data obtained from each cluster, we still observed two major branches on the dendrogram: the low biofilm producers were in a branch gathering clusters 3, 4, and 5, and the high biofilm producers were in another branch gathering clusters 1 and 2.

Genomic differences of potential biofilm associate genes presence have already been identified in *B. thermosphacta* but could not be linked to a biofilm production phenotype [22,23]. Those biofilms were grown in the optimal growth conditions of *B. thermosphacta* (i.e., BHI broth at 25 °C in polystyrene plate). Nevertheless, growth conditions have already been shown to play an important role on biofilm production and structure [54]. The decrease in temperature has been shown to increase biofilm formation of *Pseudomonas lundensis* [55], but, in contrast, it decreases the biofilm formation of *Listeria monocytogenes* [56]. The biovolume of the *L. monocytogenes* EGD-e was impacted by the medium composition [52]. Additionally, the nature of the abiotic surfaces material where the biofilm is attached seems to play a role during the first stage of the biofilm formation; *Flavobacterium psychrophilum* show a higher biofilm formation on stainless steel compared to polystyrene polyurethane and polycarbonate [57,58].

The biofilm’s characteristics provides protection against mechanical, chemical, and oxidative stresses compared to free living cells [28]. Biofilms and their 3D structure have been related to the persistence of bacteria in the food industries and their tolerance to biocide compounds, resulting in food contamination [59,60,61].

These different biofilm analysis methods provide different information about biofilm formation by *B. thermosphacta*. The cohesion, thickness, volume, surface area, and cell concentration of the biofilm could explain the presence of *B. thermosphacta* on industrial food surfaces and its persistence in food industry environments. In this study, the seven *B. thermosphacta* strains identified as high biofilm producers were isolated from seafood: three from fish, three from shrimp, and one from a salmon filleting machine. Studying more strains from even more diverse origins could provide evidence of a potential relationship with their biofilm phenotypes.

Finally, this study improves our knowledge on *B. thermosphacta* traits and its ability to adhere and colonize surfaces through biofilm formation. A study of (i) its ability to produce biofilms on various abiotic surfaces (e.g., stainless steel, polystyrene, polyurethane, and polycarbonate) typically used in the agri-food industry, (ii) its ability to produce biofilms at the low temperatures used in food processing plants, and (iii) its interactions with background biofilm microbiota would allow us to better understand its persistency within food processing environments.

## Figures and Tables

**Figure 1 microorganisms-10-02474-f001:**
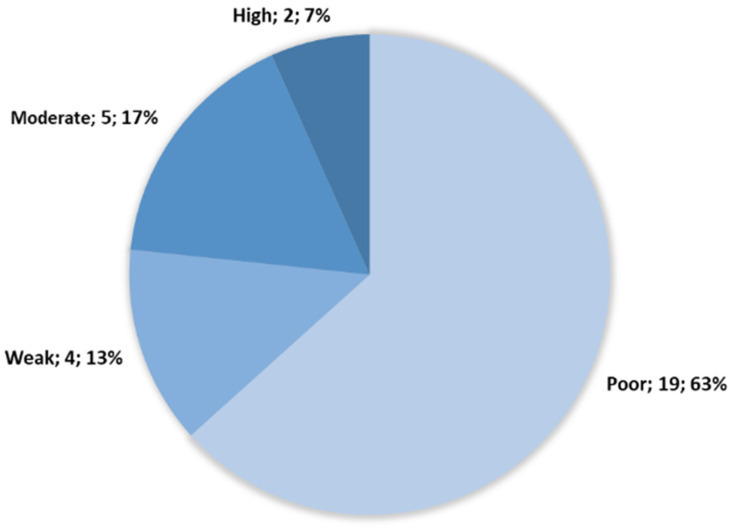
Strain distribution following cBRT clustering and size of each cluster (cluster; number of strains; %). A total of 19 strains were classified as poor early biofilm producers, representing 63% of our strains sample. A total of 4 strains as weak early biofilm producers (13%). A total of 5 as moderate early biofilm producers (17%). A total of 2 as high early biofilm producer (7%).

**Figure 2 microorganisms-10-02474-f002:**
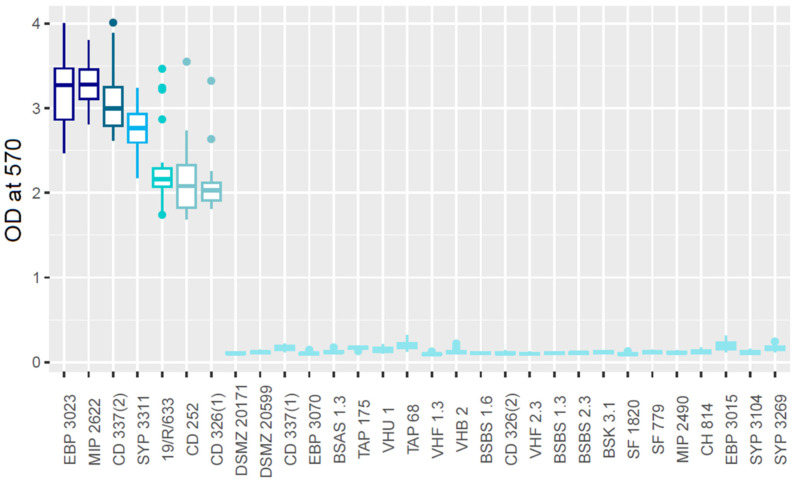
Optical density at 570 nm of biofilm producers stained with crystal violet. Each color group is statistically different from the others (*p* < 0.05). A total of 23 strains show a low mature biofilm production, while 5 show a high mature biofilm production.

**Figure 3 microorganisms-10-02474-f003:**
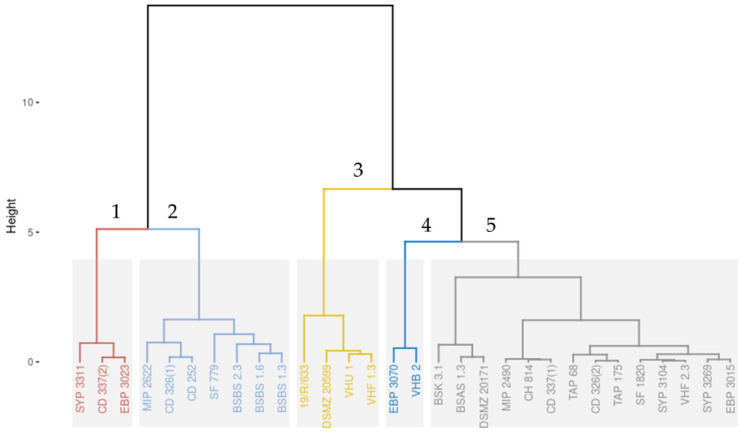
Hierarchical clustering on principal components (HCPC) dendrogram. Principal components were defined using principal component analysis (PCA) on the 74 BiofilmQ parameters extracted from confocal laser scanning microscopy (CLSM) images. Five clusters were identified based on the automatic cut tree of HCPC tool. Those clusters were numerated and colored as 1—red, 2—light blue, 3—yellow, 4—dark blue, and 5—grey.

**Figure 4 microorganisms-10-02474-f004:**
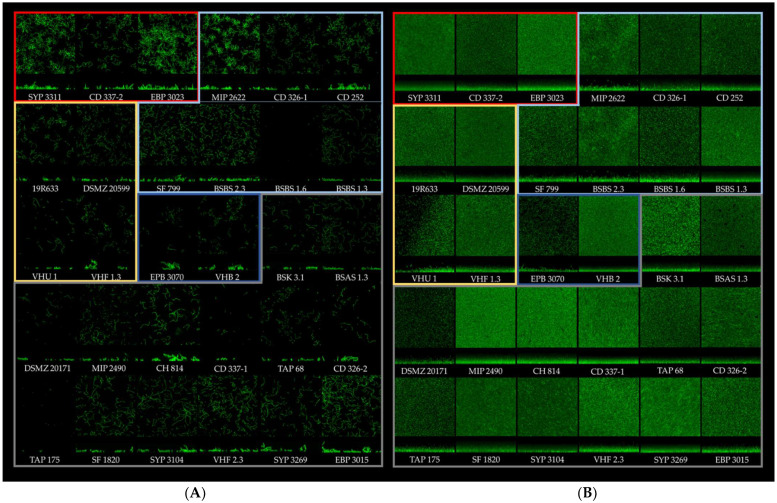
IMARIS projection from CLSM images of each strain at t = 2 h (**A**) and t = 24 h (**B**). Each cluster is framed by its clustering color code from Figure 3: red, cluster 1; light blue, cluster 2; yellow, cluster 3; dark blue, cluster 4; gray, cluster 5.

**Table 1 microorganisms-10-02474-t001:** Thirty *B. thermosphacta* strains from multiple origins.

Scheme.	Ecological Source	Strain Origin	References
DSMZ^T^20171	Fresh pork sausage	DSM/ATCC	[44]
DSMZ20599	Bacon	DSM/ATCC	
CD 337(1)	Peeled shrimp	INRAE-SECALIM	[45]
CD 337(2)	Peeled shrimp	INRAE-SECALIM	
CD326 (1)	Peeled shrimp	INRAE-SECALIM	
CD326 (2)	Peeled shrimp	INRAE-SECALIM	
CD252	Peeled shrimp	INRAE-SECALIM	
EBP3070	Smoked salmon	INRAE-SECALIM	[16]
MIP2622	Salmon	INRAE-SECALIM	
MIP2490	Salmon	INRAE-SECALIM	
BSAS1.3	Bovine slaughterhouse animal skin	INRAE-SECALIM	
VHU1	Chopped steak	INRAE-SECALIM	
VHF1.3	Chopped steak	INRAE-SECALIM	
VHB2	Butcher’s chopped steak	INRAE-SECALIM	
VHF2.3	Chopped steak-Férial	INRAE-SECALIM	
TAP175	Chicken thigh (MAP)	INRAE-SECALIM	
TAP68	Chicken thigh (MAP)	INRAE-SECALIM	
BSBS1.6	Beef slaughterhouse environment	INRAE-SECALIM	
BSBS1.3	Beef slaughterhouse environment	INRAE-SECALIM	
BSBS2.3	Beef slaughterhouse environment	INRAE-SECALIM	
BSK3.1	Bovine slaughterhouse knife	INRAE-SECALIM	
SF1820	Smoked salmon	INRAE-SECALIM	[17]
SF779	Smoked salmon	INRAE-SECALIM	
19/R/633	Salmon-filleting machine after cleaning	INRAE-SECALIM	
CH814	Saint Nectaire cheese	UMRF Aurillac-France	[3]
EBP3015	Cod (MAP)	IFREMER Nantes	[46]
EBP3023	Cod (MAP)	IFREMER Nantes	
SYP3104	Fresh tuna	IFREMER Nantes	
SYP3269	Fresh red drum	IFREMER Nantes	
SYP3311	Red drum (MAP)	IFREMER Nantes	

MAP: modified atmosphere packaging.

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
