# Peer review of "Exploring the Diversity of Biofilm Formation by the Food Spoiler Brochothrix thermosphacta"

_microorganisms, 2022, doi:10.3390/microorganisms10122474_

Round 1
Reviewer 1 Report
COMMENTS TO THE MANUSCRIPT “Exploring the diversity of biofilm formation by the food spoiler Brochothrix thermosphacta” by Gaillac et al.
General comment:
The submitted manuscript analyzes the intraspecific variation in biofilm formation between 30 strains of the food spoiler bacteria Brochothrix thermosphacta. The early biofilm formation of the strains was determined by biofilm ring test whereas the mature biofilm was analyzed by crystal violet staining. The structure of the biofilm was characterized by confocal laser scanning microscopy. The authors identified two main groups within the studied strains, the low and the high biofilm producers. The seven strains of the high biofilm producers group come from food sea samples. The biofilm of this group presents a flat architecture, high surface coverage and area, and high cell volume. The authors states that this is the first study on biofilm heterogeneity in B. thermosphacta and foresee the realization of further studies on biofilm formation of this species on different kind of surfaces and its interaction with the background microbiome to better understand its persistence in the food chain environments. All manuscript sections are clearly explained, and the obtained results are properly described and discussed. The subject of the submitted manuscript is of interest for microbiologists and food quality research. The manuscript is suitable to be published in the Microorganisms Journal. Below are some specific comments for the authors' consideration.
Specific comments and suggestions:
1. As you correctly states, there is a lack of international standard of biofilm formation assays (lane 52). However, several bioreactors for biofilm analysis have been developed for clinically relevant bacterial pathogens with its corresponding standard assays stablished by American for Testing and Material (ASTM) (Gomes et al 2019. Critical Rev Biotechnol.38: 657-670). One of these devices (Calgary device) has been used to analyze biofilm formation of food spoilage bacteria (Røder et al 2015. Food Microbiol. 51: 18-24). I think is relevant to include such standardization efforts antecedent in your introduction section.
2. Please use the plural acronym “Candida spp.” instead the singular “Candida sp.”, without italics in the acronym (spp.) (line 272). By other side, in relation with citing this Candida spp. reference, how relevant is to compare fungal (eukaryotic) with bacteria biofilm formation capabilities? The wide structural differences in cell wall and cell surface makes difficult to compare its biofilms.
3. The biofilm formation experiments were conducted at 25 °C, but meat and food sea are preserved at low temperature, commonly at 4 °C, in the food chain before consumption. It has been demonstrated that biofilm formation by food spoilage bacteria, including B. thermosphacta, varies depending on the temperature (Flemming et al 2022. Nat Rev Microbiol. https://doi.org/10.1038/s41579-022-00791-0; Chen et al 2020. Can J Microbiol. 66: 303–312). Can you please comment and discuss on this regard?
4. It has been described two main B. thermosphacta cell morphologies (I and II) that originates loose and compact clumps (Chen et al 2020. Can J Microbiol. 66: 303–312). Can you visualize in your microphotographs any cell similarities with those previously documented? Can these cell morphologies help to explain the differences in biofilm formation of your strains? I think it is relevant to consider the inclusion of the Chen et al. (2020) work in your discussion. Further, this previous work, despite do not use biofilm analysis techniques as used your work, contradicts your statement that the submitted manuscript “…is the first to describe the diversity of B. thermosphacta biofilm formation.”, because it describes two main cell aggregation types (biofilm) and other two subtypes in 24different strains.
5. Besides the cell morphology, what other variables can explain the differences in biofilm formation between studied strains? Please provide possible explanations on components of the biofilm matrix (exopolysaccharides, proteins, DNA) (Flemming et al 2022. Nat Rev Microbiol. https://doi.org/10.1038/s41579-022-00791-0), that can aid to explain the observed results and can be analyzed in further studies. Furthermore, a recent study cited by you state that B. thermosphacta genome carries only one biofilm associated gene (lmaC)but showed the set of genes for cellulose production, a biofilm associated component(Wagner et al 2021. Int J Food Microbiol. 349: Article ID 109232). Also, the biofilm of one B. thermosphacta strain in the same reference showed low levels of carbohydrates and the absence of protein and extracellular DNA (eDNA). All these antecedents can aid to improve your discussion in seeking for probable explanations of the obtained results.
Author Response
Dear reviewer,
We thank you very much for your advice review of our manuscript and your relevant comments. We tried to take into account all your comments to revise our manuscript. Hoping it will be consistent to your expectation.
Sincerely.
Pr. Hervé Prévost,
Corresponding author.
- As you correctly states, there is a lack of international standard of biofilm formation assays (lane 52). However, several bioreactors for biofilm analysis have been developed for clinically relevant bacterial pathogens with its corresponding standard assays stablished by American for Testing and Material (ASTM) (Gomes et al 2019. Critical Rev Biotechnol.38: 657-670). One of these devices (Calgary device) has been used to analyze biofilm formation of food spoilage bacteria (Røder et al 2015. Food Microbiol. 51: 18-24). I think is relevant to include such standardization efforts antecedent in your introduction section.
Response
As you said it’s relevant to include the effort of standardization of biofilm formation assays in the introduction. As a CV staining derived method, Calgary device was included with the CV methods description and we propose some modification in the manuscript to highlight biofilm quantification standardization effort. As very pertinent references Gomes et al 2019 and Røder et al 2015 were added respectively [31] and [37].
Modifications in the manuscript:
Page 2, line 57 “Numerous complementary methods have been developed for in vitro biofilm growth and quantification, but without any international standardization to date [29,30]. Biofilms reactor devices have been developed for bacterial biofilm culture with established standard assays by the American for Testing and Material [31]. The crystal violet (CV) staining methods in microplates and its variant were widely used to quantify fixed biomass. CV methods were adapted and improved from various microorganisms including human pathogens [32–36] and food processing associated microorganisms [37,38].”
- Please use the plural acronym “Candida spp.” instead the singular “Candida sp.”, without italics in the acronym (spp.) (line 272). By other side, in relation with citing this Candida spp. reference, how relevant is to compare fungal (eukaryotic) with bacteria biofilm formation capabilities? The wide structural differences in cell wall and cell surface makes difficult to compare its biofilms.
Response
You’re right those characteristics makes difficult to compare their biofilms. To clarify the discussion, we deleted the sentence with “Candida spp.”
- The biofilm formation experiments were conducted at 25 °C, but meat and food sea are preserved at low temperature, commonly at 4 °C, in the food chain before consumption. It has been demonstrated that biofilm formation by food spoilage bacteria, including B. thermosphacta, varies depending on the temperature (Flemming et al 2022. Nat Rev Microbiol. https://doi.org/10.1038/s41579-022-00791-0 ; Chen et al 2020. Can J Microbiol. 66: 303–312). Can you please comment and discuss on this regard?
Response
It is true, the temperature, medium and the abiotic surface used in this study will have an impact on the biofilm formation and structure. We comment in the discussion the impact of growth conditions on biofilm characteristics. Were added some references citations Flemming et al 2022, Liu et al 2015, Pan et al 2009, Vidal et al 2020 and Abdallah et al 2014, respectively [54] to [58].
Modifications in the discussion part:
Page 10, line 344. “Those biofilms were grown in the optimal growth conditions of B. thermosphacta (i.e. BHI broth at 25°C, in polystyrene plate). Nevertheless, growth conditions have already been described to play an important role on biofilm production and structure [54]. The decrease of temperature has been shown to increase biofilm formation of Pseudomonas lundensis [55] but in contrast decreases the biofilm formation of Listeria monocytogenes [56]. The biovolume of the L. monocytogenes EGD-e was impacted by the medium composition [52]. Also, the nature of the abiotic surfaces material where the biofilm is attached seems to play a role during the first stage of the biofilm formation, Flavobacterium psychrophilum show a higher biofilm formation on stainless steel compared to polystyrene polyurethane, and polycarbonate [57,58].”
- 4. It has been described two main thermosphactacell morphologies (I and II) that originates loose and compact clumps (Chen et al 2020. Can J Microbiol. 66: 303–312). Can you visualize in your microphotographs any cell similarities with those previously documented? Can these cell morphologies help to explain the differences in biofilm formation of your strains? I think it is relevant to consider the inclusion of the Chen et al. (2020) work in your discussion. Further, this previous work, despite do not use biofilm analysis techniques as used your work, contradicts your statement that the submitted manuscript “…is the first to describe the diversity of B. thermosphactabiofilm formation.”, because it describes two main cell aggregation types (biofilm) and other two subtypes in 24 different strains.
Response
None of those cell morphologies were observed during our study.
In this article, we consider the definition of biofilm as a microbial community attached to a surface and embedded in a self-produced extracellular matrix. According to Chen et al 2020, those aggregations do not present any extracellular matrix and “gentle swirling of the plate released the cell mat from the bottom of the plate, resulting in a mostly intact, free-floating mesh or film in a culture broth with little or no visible turbidity”. As described by Chen et al 2020, those cells morphologies were considered as an alternative mode of biofilm formation not as a biofilm. In order to clarify we mentioned that our work is to our knowledge the first to be published on the characterization of Brochothrix thermosphacta biofilms on abiotic surface. The paper by Chen et al 2020 [25] is cited in the introduction.
Modification in the manuscript:
Page 2, line 46 “Moreover, two cell morphology was identified, type I (BI-like fluffy cloud-like clumps) and type II (BII-like dense clusters with protrusions), those morphotype may influence cell aggregation and biofilm formation [25]”
Page 2, line 48 “However, to our knowledge, no study on the biofilm-forming ability on abiotic surface with so many different strains of B. thermosphacta, has been published yet.”
Page 2, line 90 “To our knowledge, this study is the first to describe the diversity of B. thermosphacta biofilm formation with a such set of complementary biofilm assays on abiotic surface.”
- Besides the cell morphology, what other variables can explain the differences in biofilm formation between studied strains?
Please provide possible explanations on components of the biofilm matrix (exopolysaccharides, proteins, DNA) (Flemming et al 2022. Nat Rev Microbiol. https://doi.org/10.1038/s41579-022-00791-0), that can aid to explain the observed results and can be analyzed in further studies. Furthermore, a recent study cited by you state that B. thermosphacta genome carries only one biofilm associated gene (lmaC) but showed the set of genes for cellulose production, a biofilm associated component (Wagner et al 2021. Int J Food Microbiol. 349: Article ID 109232). Also, the biofilm of one B. thermosphacta strain in the same reference showed low levels of carbohydrates and the absence of protein and extracellular DNA (eDNA). All these antecedents can aid to improve your discussion in seeking for probable explanations of the obtained results.
Response
In the paper by Wagner et al 2021. B. thermosphacta genome carries only one biofilm associated gene (lmaC) (Table 2). As you highlight, cellulose production associate genes were identified in genomic bank of B. thermosphacta genomes. Biofilm associate gene investigation is needed for B. thermosphacta high and low biofilm producer and could be explored in a further study. Was cited in the introduction Solomon et al 2005 [24].
In order to take into account this comment the discussion part was modified as follow:
Page 1, line 42 “Genes involved in cellulose production were identified in B. thermosphacta genomes [22]. Cellulose production genes were already associated to biofilm matrix production in Salmonella enterica [24]”
Page 10, line 340 “The differences of biofilm production between strains could be explained by a difference of extracellular matrix component production, extracellular matrix component degradation, presence of adhesin [28]. Genomic differences of potential biofilm associate genes presence has already been identified in B. thermosphacta but could not be linked to a biofilm production phenotype [22,23].”
Reviewer 2 Report
Title
The title and the aim of the study are clearly constructed.
Abstract
The abstract includes the aim of the study, methods used in the experiment and contain the principal results and conclusions.
Introduction
The author did not provide a clear explanation of the research gap and the purpose of the study in the introduction part.
Methods
The data is well collected. The methods are described in detail, in the way which permits the research to be replicated. The sampling is appropriate and adequately described.
Results
The results were discussed extensively, in a clear and legible way.
Discussion
They correctly interpreted and described the significance of the results for the research. They skillfully referred to the results of other researchers.
Language
The article needs to be proofreading, lots of typos and mistakes that resulting in a low flow of information. Most of the references are not up-to-dated correctly written.
Author Response
Dear reviewer,
We thank you very much for your advice review of our manuscript and your relevant comments. We tried to take into account all your comments to revise our manuscript. Hoping it will be consistent to your expectation.
Sincerely.
Pr. Hervé Prévost,
Corresponding author.
- Introduction: The author did not provide a clear explanation of the research gap and the purpose of the study in the introduction part.
Response:
We clarify the research gap and the purpose of the study.
Modifications in the manuscript:
Page 2, line 84: “This article explores the production capability and structure diversity of B. thermosphacta biofilms. This is important to improve our knowledge on biofilm behaviour in food processing environment and better control food contamination by this bacteria.”
- Language. The article needs to be proofreading, lots of typos and mistakes that resulting in a low flow of information. Most of the references are not up-to-dated correctly written.
Response:
To correct the typos and mistakes, we checked all the manuscript and references. Furthermore, the manuscript has been proof read by a certified English language-editing service to meet the international English language standards for publication, Annie Buchwalter (https://anniebuchwalter.wixsite.com/monsite/prestations), professional proof-reader and translator at the French National Research Institute for Agriculture, Food and Environment (INRAE).
Reviewer 3 Report
1. Figure 1 is very poor, no proper explanation or interpretation in the footnote.
2. Most of the figure’s resolution are very less and poor, please improve the footnote as well
3. The putative O-acetyltransferase can be estimated through rt-PCR?
4. Author must check the house keeping genes of different strains to measure the biofilm quantitatively
5. The discussion on mode of the biofilm and planktonic lifestyle of bacteria is missing
6. Please add a note on how these biofilm life style contributes to food spoilage
7. I suggest performing an experiment on biofilm formation ability on food surface with respect to poor and high biofilm forming bacteria and quantify the same.
8. Please check for typo error and spelling throughout the manuscript.
Author Response
Dear reviewer,
We thank you very much for your advice review of our manuscript and your relevant comments.
We tried to take into account all your comments to revise our manuscript. Hoping it will be
consistent to your expectation.
Sincerely.
Pr. Hervé Prévost,
Corresponding author.
- Figure 1 is very poor, no proper explanation or interpretation in the footnote.
- Most of the figure’s resolution are very less and poor, please improve the footnote as well
Response
The resolution of the images was modified and their footnote were improved to better defined and explained their purpose.
Modifications in the manuscript:
Page 6, line 212 “Figure 1. Strain distribution following cBRT clustering, and size of each cluster (cluster; number of strains; %). 19 strains were classified as poor early biofilm producers, representing 63% of our strains sample. 4 strains as weak early biofilm producers (13%). 5 as moderate early biofilm producers (17%). 2 as high early biofilm producer (7%).”
Page 7, line 226 “Figure 2. Optical density at 570 nm of biofilm producers stained with crystal violet. Each color group is statistically different from the others (p<0.05). 23 strains shown a low mature biofilm production while 5 shown a high mature biofilm production.”
Page 7, line 241 “Hierarchical Clustering on Principal Components (HCPC) dendrogram. Principal Components were defined using Principal Component Analysis (PCA) on the 74 BiofilmQ parameters extracted from Confocal Laser Scanning Microscopy (CLSM) images. Five clusters were identify based on the automatic cut tree of HCPC tool. Those clusters were numerated and coloured as 1-red, 2-light blue, 3-yellow, 4-dark blue and 5-grey.”
- The putative O-acetyltransferase can be estimated through rt-PCR?
- Author must check the house keeping genes of different strains to measure the biofilm quantitatively
Response
The putative O-acetyltransferase gene has been identified in a previous study (Illikoud et al 2018) and use in our manuscript (introduction) only to highlight the potential biofilm production capability by B. thermosphacta. Its expression could be indeed quantified by rt-PCR in a next study, in association to biofilm production phenotype and a genomic comparison, after a whole genome sequencing of the high and low biofilm producers. To normalized this O-acetyltransferase quantification, housekeeping genes are effectively needed, as it has been explored and explained by (Nailis et la. 2006, PMID: 16889665). This strategy seems to be relevant and will be possibly considered in a next study.
- The discussion on mode of the biofilm and planktonic lifestyle of bacteria is missing
- Please add a note on how these biofilm life style contributes to food spoilage
Response
We took into account these elements in the discussion of the manuscript by adding the following sentences.
Modifications in the manuscript:
Page 11, line 352 “Biofilm life style provide protection against mechanical, chemical and oxidative stresses compered to free living cells [28]. Biofilms and their 3D structure have been related to the persistence of bacteria in the food industries and their tolerance to biocide compounds, resulting in food contamination [59–61].”
- I suggest performing an experiment on biofilm formation ability on food surface with respect to poor and high biofilm forming bacteria and quantify the same.
Response
Impact of surfaces materials and medium composition on biofilm formation were well described, we propose at the end of the discussion to go further
- its ability to produce biofilms on various abiotic surfaces (e.g. stainless steel, poly-styrene, polyurethane, polycarbonate) typically used in the food industry,
- its ability to produce biofilm at low temperature used in food processing plants and
- its inter-actions with background biofilm microbiota would allow us to better understand its persistence within food-processing environments.
Modifications in the manuscript:
Page 11, line 342 “Those biofilms were grown in the optimal growth conditions of B. thermosphacta (i.e. BHI broth at 25°C, in polystyrene plate). Nevertheless, growth conditions have already been described to play an important role on biofilm production and structure [54]. The decrease of temperature has been shown to increase biofilm formation of Pseudomonas lundensis [55] but in contrast decreases the biofilm formation of Listeria monocytogenes [56]. The biovolume of the L. monocytogenes EGD-e was impacted by the medium composition [52]. Also, the nature of the abiotic surfaces material where the biofilm is attached seems to play a role during the first stage of the biofilm formation, Flavobacterium psychrophilum show a higher biofilm formation on stainless steel compared to polystyrene polyurethane, and polycarbonate [57,58].”
- Please check for typo error and spelling throughout the manuscript.
Response
To correct the typos and mistakes, we checked all the manuscript and references. Furthermore, the manuscript has been proof read by a certified English language-editing service to meet the international English language standards for publication, Annie Buchwalter (https://anniebuchwalter.wixsite.com/monsite/prestations), professional proof-reader and translator at the French National Research Institute for Agriculture, Food and Environment (INRAE).
Submission Date
25 November 2022
Date of this review
06 Dec 2022 08:38:15
Round 2
Reviewer 3 Report
All of the review comments are included in the revision